# Patient preferences for diagnostic imaging services: Decentralize or not?

Eline M. van den Broek-Altenburg[1]*, Jamie S. Benson[2], Adam J. Atherly[3], Kristen K. DeStigter[1]

1 Larner College of Medicine, University of Vermont, Burlington, USA, 2 Perelman School of Medicine, University of Pennsylvania, Philadelphia, USA 3 College of Health Professions, Virginia Commonwealth University, Richmond, USA

* Eline.altenburg@med.uvm.edu

## Abstract

The objective of this study was to identify patient preferences for outpatient diagnostic imaging services and analyze how patients make trade-offs between attributes of these services using a discrete choice experiment (DCE). We used a DCE with 14 choice questions asking which imaging locations patients would prefer. We used latent class analysis to analyze preference heterogeneity between different patient groups and to estimate the relative value they assign to different attributes of imaging services. Our analysis showed that the "Experienced Patients" subgroup generally value diagnostic imaging services in both acute and chronic situations and had a strong preference for hospital outpatient radiology departments (HORD) that would provide services at lower costs, where their images would be interpreted by a specialty radiologist, the clinic would be recommended by their PCP, online scheduling would be available, service rating were higher, and travel and wait times would be shorter. New Patients significantly valued the service rating of the (HORD and online scheduling. HORDs can be more competitive by providing services that live up to expectations better than available retail radiology clinics (RRCs). Most RRCs do not currently offer online scheduling so ease of use may also steer patients towards HORDs. HORDs have the advantage of being linked to the main medical center which has the reputation of having clinical expertise and more sophisticated technology. We conclude that there is room for medical centers to build HORDs that provide an appealing and competitive alternative to current RRC.

## Introduction

An important challenge for many hospitals and clinics is addressing the escalating diagnostic imaging demands for hospital outpatient radiology departments (HORDs) [1]. HORDs have struggled to keep up with the increasing need for imaging, with MRI and CT imaging rates for adults more than doubling from 2006 to 2016 [2]. This

**Data availability statement:** All relevant data for this study are publicly available from the GitHub repository (https://github.com/ElinevandenBroek/DecentralizedImaging).

**Funding:** This was an individual's research award from the PhRMA Foundation. The funders had no role in study design, data collection and analysis, decision to publish, or preparation of the manuscript.

**Competing interests:** The authors have declared that no competing interests exist.

escalating demand for services, coupled with an increase in patient complexity not only leads to increased costs, but also reduced accessibility and quality. Delays in imaging at HORDs can also lead to delayed care delivery, which can negatively impact patient outcomes.

Retail radiology clinics (RRCs) emerged from the general retail health clinics' market boom as a potential solution to the imaging bottleneck effect of HORDs. RRC offer a variety of diagnostic imaging services, typically at lower cost, at a decentralized clinic [3–5]. RRCs have gained popularity by cutting wait-times, increasing accessibility, and reducing costs [3–5]. Despite the increase in RRC utilization, price variability and unreliable quality of the technology used, imaging interpretation and reporting of images have led RRCs to be labeled as low-value substitutes to HORD services [6–8]. For example, one study found significant variability in diagnostic reports from 10 different RRCs performing an MRI scan of the lower back on the same patient within a 3-week period [5]. Another study reported that radiology departments in tertiary care centers are frequently asked to perform secondary interpretations of imaging studies, finding that discrepancy rates vary widely [6].

In parallel, hospital networks are struggling with care integration and coordination, as well as maintaining quality standards across all departments and campuses within their network. Some hospital systems are now looking at options to develop their own RRCs to provide convenient and efficient locations, without sacrificing the technological or interpretation quality, while leveraging the existing trust patients may have with their hospital system [3–4]. This is a potentially valuable innovation for patients, as they would be able to receive the "specialty read" quality at locations with shorter wait times and lower out-of-pocket costs.

It remains unclear, however, what aspects of imaging services patients value the most. In the push for patient-centeredness, the Centers for Medicare and Medicaid Services (CMS) recommends hospital departments incorporate patient insight into their service quality measures [7]. Despite widespread encouragement to incorporate the patient voice in health delivery decisions, developments such as RRCs are usually done without meaningful input from patients [8]. Existing research on patient preferences in diagnostic imaging services is sparse and what exists is often contradictory [9–15].

The few studies that did investigate imaging preferences focused largely on reporting methods. One study looked at patient preferences for CT and MRI imaging results and focused on how, from whom, and how soon patients preferred the results [10]. The authors used a survey and asked those questions directly of patients. They found that patients wanted their results to be communicated much sooner than is currently practiced. Another study focused on receiving radiology results and reviewing the images and findings directly with a radiologist after completion of an examination [11]. Patients preferred hearing examination results from both their ordering provider and the interpreting radiologist [12]. Other than results communication, there was a study focused on the referral mechanism [13] and the importance of wait time for test results [14]. None of these studies focused on patient preferences for *where* to seek imaging services, including practical considerations such as parking and logistics.

There have been studies focused on preferences for primary care services [15] and preferences for health service delivery in general [16], but this study will specifically focus on preferences for imaging services.

The objective of this study was to identify patient preferences for outpatient diagnostic imaging services and analyze how patients make trade-offs between attributes of these services using a stated choice experiment. In this study, we analyze patient's individual preference heterogeneity for these services and study how preferences vary among and within different patient populations.

## Methods

### Qualitative study

To elicit stated preferences for imaging services, we designed a Discrete Choice Experiment (DCE) which allows researchers to analyze the trade-offs that patients are willing to make, including options that may not currently exist but could in the future [17]. Before the experimental design, we needed to explore which attributes of imaging services matter to patients. We therefore conducted focus groups among patients at a middle-size academic medical center who volunteered to share their experiences with imaging services. (henceforth referred to as the medical center). Inclusion criteria consisted of 1) adults over 18 years who 2) have received outpatient radiology imaging services in the last year. Exclusion criteria consisted of 1) persons under the age of 18 with 2) inpatient status or 3) not having received outpatient radiology services in the last year. Participants were fully informed and gave their consent by participation in the focus group and demographic surveys. We recruited from a pre-existing group of patient advisors to the medical center. Prior to the session, participants were provided a detailed information sheet on the project. Compensation for participation included a meal and parking vouchers.

Two focus groups with 12 participants total were conducted in a semi-structured manner lasting between 90 and 120 minutes. Group sizes were intended to fall between 4 and 8 people in order to stimulate a conducive group conversation without the risk of too many voices for individual experiences to be shared [18]. A trained member of the project moderated the focus groups using a question guide featuring questions aimed at understanding how patients perceive their radiology experiences led participants through their focus group session. Questions were intended to be exploratory and sometimes followed by probes to allow differences between patient insights and experiences to emerge. All sessions were audio recorded with participants' informed consent. Member checking assessed the credibility of responses, whereby the moderator paraphrased their interpretation of an ambiguous response followed by participants' confirmation or rephrasing [19]. Follow-up probes asking for more detail or specific examples were also used. The demographic characteristics of the participants can be found in the Appendix 2.

We transcribed the focus groups following their completion. The transcriptions were later analyzed using ATLAS.ti version 8 qualitative analytic software, following thematic analysis technique of phase 1) familiarization with the data through reading, re-reading and noting initial ideas, 2) generation of initial codes, 3) searching for potential themes and sequestering all relevant data to the theme, 4) reviewing themes and generating a thematic map, and lastly 5) refining and specifying themes before producing the final report [20]. Themes are reported based on their frequency within groups and across groups, and intensity at which themes were discussed. The frequencies of the participant-identified attributes discussed within and across focus groups are outlined in Table 1 to illustrate how the attributes in the DCE were defined.

### Discrete choice experiment

Following the focus groups, we designed a DCE in which patients were asked to choose the imaging clinic they preferred. Each choice task had three different clinics which varied by 10 different attributes, which were determined by the themes identified in the focus groups. Four attributes had 2 levels, and six attributes had 3 levels. The levels were partly defined on concrete contributions from patients in the focus groups and partly on information from the medical center around realistic real-life levels.

**Table 1. Participant-identified attributes discussed frequency by focus group.**

| PARTICIPANT-IDENTIFIED ATTRIBUTES | Focus Group 1 | | | Focus Group 2 | | | Totals |
| --- | --- | --- | --- | --- | --- | --- | --- |
| | Absolute[1] | Relative % across[2] | Relative % within[3] | Absolute[1] | Relative % across[2] | Relative % within[3] | Absolute[1] |
| *Staff attentiveness/ bedside manner* | 25 | 0.38 | 0.18 | 41 | 0.62 | 0.30 | 66 |
| *Results explained by knowledgeable staff* | 25 | 0.45 | 0.18 | 30 | 0.55 | 0.22 | 55 |
| *Communicated time to expect results* | 29 | 0.67 | 0.21 | 14 | 0.33 | 0.10 | 43 |
| *Patient-centered scheduling* | 22 | 0.65 | 0.16 | 12 | 0.35 | 0.09 | 34 |
| *Parking accessibility* | 9 | 0.45 | 0.06 | 11 | 0.55 | 0.08 | 20 |
| *Information on procedure* | 10 | 0.45 | 0.07 | 12 | 0.55 | 0.09 | 22 |
| *Facility's reputation* | 20 | 0.53 | 0.14 | 18 | 0.47 | 0.13 | 38 |
| Totals | 140 | 0.50 | 1.00 | 138 | 0.50 | 1.00 | 278 |

[1]The absolute number of references to or comments regarding the associated attribute by participants

[2]The relative frequency (%) of the attribute compared across focus groups

[3]The relative frequency (%) of the attribute compared to all other attributes within the focus group

Table 2 shows the attributes and levels used in the DCE. Following the analysis of the focus groups, we included the following attributes: whether the interpreting radiologist is a general or sub-specialty radiologist; whether the clinic was recommended by their primary care physician (PCP); time to results; out of pocket cost; wait time to an appointment; travel time to the clinic; parking costs; parking accessibility; service; and whether or not online scheduling is available. Service is a multifactorial attribute (e.g., staff attentiveness and facility amenities) combined into a star rating. The rating scale is between one and five stars, with a five-star rating representing an excellent service and a one-star rating suggesting a poor service, as rated by other hypothetical patients. The star ratings are based on the CMS Five-Star Quality Rating System which were created to help consumers, their families, and caregivers compare clinics more easily and to help identify areas about which they may want to ask questions. A rating of 1 or 2 stars means that the clinic's performance was below the average of other agencies on selected measures; it does not necessarily mean care is poor. A rating of 4 or 5 stars means that the clinic's performance was above the average of other agencies on selected measures. Costs were "pivoted" around a respondent's current out of pocket costs: $25 less or $25 more. We used pivot style stated choice data for OOP costs to include a reference alternative whose attributes remain invariant across replications for the same respondent [21].

Participants were asked to imagine a situation where they were hurt and needed imaging services. In the survey, the following descriptions of the choice situation were given based on common reasons for imaging services:

- Situation 1):"For the purpose of this study, suppose you hurt your arm and your primary care provider wants to send you for an X-ray. You have three options of locations where you can have your imaging done."

- Situation 2):"For the purpose of this study, suppose you hurt your back a while ago and are having persistent pain. Your primary care provider wants to send you for an MRI. You have three options of locations where you can have your imaging done."

Following the DCE choice tasks, we asked patients attitudinal questions on a 1–5-point Likert scale from "strongly disagree" to "strongly agree". These questions were borrowed from and validated by the national Medical Expenditures Panel Survey (MEPS) [19], a set of large-scale surveys of families and individuals, their medical providers, and employers

**Table 2. Overview of all the attributes and levels.**

| Attribute | Level 1 | Level 2 | Level 3 |
|---|---|---|---|
| Interpreting Doctor Specialty | General Radiologist – This means that your images would be interpreted by a Radiologist, which is a doctor who is a specialist in interpreting multiple types of images but does not have specific additional training in the type of image you're getting. (For example, the doctor interprets X-Rays but also interprets CT scans, ultrasounds, and other images of multiple body systems) | Specialty Radiologist – This means that your images would be interpreted by a Radiologist who has additional training interpreting the type of image you're getting. (for example, the doctor interpreting your arm X-Ray is specialized in reading images of broken bones in the arm). | |
| Primay Care Recommendation | Yes – This means that the clinic is recommended by your primary care provider | No – This means that the clinic is not specifically recommended by your primary care provider | |
| Wait Time to Results | 25 minutes less - This would mean that the clinic will take 25 minutes less to give you the results from your x-ray | Same as current source of care | 25 minutes more |
| Costs | 25 $ less | Same as current | 25 $ more |
| Travel Time | 25 minutes less - This would mean that the clinic is 25 minutes closer | Same as current | 25 minutes more |
| Wait Time to Appointment | 25 minutes less | Same as current | 25 minutes more |
| Parking Costs | Free – This means that the clinic is free to park at | Paid – This means that you would need to pay to park your car at the clinic | |
| Parking Accessibility | 25 minutes less - This would mean that it takes you 25 minutes less to travel from your vehicle or bus stop to the clinic waiting room | Same as current | 25 minutes more |
| Service | Less 1 - This would mean that the clinic is rated 1 star rating less for patient experience, safety, and the quality of your care. | Same as current | Plus 1 |
| Online Scheduling | Not Available – This means that online scheduling is not available at the clinic | Available – This means that online scheduling is available at the clinic | |

across the United States. The questions focused on perceived need for healthcare, need for insurance, risk-aversion, and perceived personal health status.

## Experimental design

Once the relevant attributes and levels were chosen, it was desirable to exclude the dominant options, repeats and implausible attribute combinations. There are three ways to reduce the dimensions of the full factorial design matrix to fractional factorial designs: random designs, orthogonal designs, and efficient designs [17]. A design is considered more efficient if it can produce more efficient data in the sense that more reliable parameter estimates can be achieved with an equal or lower sample size [22]. The researcher specifies utility functions that include these "priors", and these are used to determine the logit probabilities, and the log likelihood functions [23,24] .

Our experimental design of the DCE was based on "prior" estimates of the utilities for attributes of the choice which were calculated by using expectations on what the model parameters will be. These numbers were first based on a mid-size medical center's experience with the length of wait time, parking opportunity, et cetera, based on administrative data and results from the focus groups. We then produced sample data from a pilot which included the first 20 respondents to the survey. We stopped data collection after their responses and estimated preliminary models based on their responses, which we refer to as "prior" estimates. Once the "prior" estimates are established and the utility functions were defined, we used software program NGene to include the specific priors from the initial 20 respondents in the utility functions. The advantage of the Ngene algorithm is that it searches for a list of choice sets in which dominant alternatives do not appear, choice sets are not repeated, and the number of choice sets for which the answer can be inferred from the previous one is minimized.

In this way, 14 choice questions in our survey were enough to derive efficient data regarding the utilities that respondents assigned to the different attributes of imaging services. In some cases, an efficient design includes many choice tasks. A blocking experimental design can then be used to avoid too much of a cognitive burden for the respondent. Blocks are subsets of the choice questions, which are usually equally sized, that contain a limited number of choice questions for each respondent. In those cases, respondents are randomly assigned to a block and answer the choice questions in that block instead of the entire design. In our study, respondents were randomized to either 14 choice questions related to X-ray services, or 14 questions related to MR services. The vignette, which can be found in Appendix 1, was different for the two DCE's and some of the levels, such as costs, were also different. We did not use overlaps since there was no significant relationship between the different attributes, such as in DCE instruments based on the EQ-5D-5L [25]. An example of a choice question regarding imaging services is shown in Fig 1. We used online research software SurveyEngine for the entire online survey, which can be found in Appendix 3, including the DCE.

### Data source, participants and study size estimation

Our data were sampled from an online Centiment panel between April 11, 2021, through November 19, 2021, with a second sample between May 16, 2022, and June 24, 2022. Centiment is a survey company which recruits individuals to answer surveys to generate rewards for themselves or to pledge their earnings to a nonprofit of their choice and it is open to anyone to participate. Centiment has engineered complex systems to manage their respondents and ensure they are providing thoughtful responses. Centiment contacted 472 individuals in the catchment area of a moderate size academic medical center in a rural Northeastern part of the United States. Participants completed written consent before continuing to the online survey questions. Without the online consent, respondents were not able to proceed. The answers were recorded in the data. All data were fully anonymized before shared with the study team. The Institutional Review Board at the University of Vermont reviewed the study and determined it was exempt from full review.

| | Clinic 1 | Clinic 2 | Clinic 3 |
|---|---|---|---|
| Interpreting Radiologist | General Radiologist | Specialty Radiologist | General Radiologist |
| Primary Care Recommendation | No | No | Yes |
| Time to Results | 42 minutes | 56 minutes | 70 minutes |
| Cost | $37 | $37 | $37 |
| Wait Time | 30 minutes | 30 minutes | 30 minutes |
| Travel Time | 81 minutes | 81 minutes | 65 minutes |
| Parking Cost | Paid | Free | Paid |
| Parking Accessibility | 22.5 minutes | 37.5 minutes | 22.5 minutes |
| Service | 2 Star: ** | 2 Star: ** | 3 Star: *** |
| Online Scheduling | Available | Not Available | Available |
| Which would you choose? | ○ Clinic 1 | ○ Clinic 2 | ○ Clinic 3 |

**Fig 1. Example of a choice task in DCE.**

Of the total sample, 268 finished the survey and met initial inclusion criteria (Age >= 18): 134 were assigned to the arm X-ray group, 134 to the back MRI group. We excluded 98 subjects for failing consistency criteria, qualifying for a closed quota, failing a bot-behavior check, or failing an attention (response quality) check, leaving a final sample of 170: 84 in X-ray, 86 in MRI. The quotas that were agreed on used Census data for age by decade, gender, region, race/ethnicity, and income.

The S-efficient design we generated in NGene showed that we needed a minimum of 55 respondents, so our sample size was sufficient. Bots were identified by the RegEx program and manual review of the free text entry boxes. Typically, the bots in our sample entered nonsense or repeated the question's text into those fields. In addition, we seek to identify fraudulent data by defining a priori indicators that warranted elimination or suspicion; an approach borrowed from another study [26].

For attention, we then checked for consistency and filtered by completion time over a threshold of 7 minutes and filtered any respondents out that showed straight-lining behavior, meaning that a respondent would always pick the same response to the choice questions [27–29]. As each subject answered 14 choice tasks, we obtained an effective sample size of n = 2,380 for modeling. We used NGene 1.2.1 (ChoiceMetrics, 2018) to estimate the minimum sample size required for this study.

## Statistical methods

We used a mixed multinomial logit model (MMNL) to estimate the probability of a choice alternative being chosen, depending on the characteristics of the choice (attributes and levels) and the characteristics of the chooser [30–32]. Mixed logit models rely on using continuous statistical distributions to represent unobserved heterogeneity [33]. A mixed logit model accommodates more flexible substitution patterns, and allows for random taste variation, unrestricted substitution patterns, and correlation in unobserved factors [30,31] . Mixed logit models make it feasible to derive individual-specific estimates conditional on the observed individual choices [34].

A different approach is to use discrete rather than continuous distributions and probabilistically segmenting a sample population into different segments, such as latent class analysis (LCA) [33]. LCA explores deterministic heterogeneity by incorporating explanatory variables as multiplicative interaction terms. We used latent class analysis (LCA) in STATA 18 (StataCorp LLC, 2023) which addresses the issue of unobserved preferences of patients by probabilistically segmenting a sample population into different groups or "classes" based on a latent variable [34]. The LCL model might explain, for example, that patients who had previous imaging services are more likely to fall into the class that is more sensitive to appointment wait time, while older patients might be more likely to fall into the class that is more sensitive to PCP recommendation. Class membership is first defined by a membership function including the indicator variables, after which the utility functions of different classes can be estimated.

We used both approaches in this study to seek some understanding of the relative merits of both modeling strategies, each regarded as an advanced interpretation of discrete choice models, as others have done [35]. Both models offer alternative ways of capturing unobserved heterogeneity and other potential sources of variability in unobserved sources of utility [35].

## Results

### Descriptive results

A total of 84 patients answered questions about preferences for attributes of an X-ray; 86 people responded to the MRI choice questions. The summary statistics are reported in Table 3. On average, patients answering choice questions on X-ray tended to be female (60%), white (91%) and live in rural areas (58%); about half had private insurance (46%) and few had met their insurance deductible (21%). Patients answering the choice questions on the MRI were similar, with 67% female, 96% white, 69% in rural areas, 45% with private insurance, 22% had met their insurance deductible. Patients receiving the X-ray choice questions had had an average of 4.8 previous images while MRI patients had had 2.9.

**Table 3. Descriptive statistics.**

| | X-ray | | MRI | |
|---|---|---|---|---|
| Age (mean, sd) | 46.0 | (16.1) | 47.8 | (16.9) |
| Income (USD) (mean, sd) | $48,425 | ($40,939) | $47,444 | ($47,881) |
| Number of Previous Images (mean, sd) | 4.8 | (1.5) | 2.9 | (1.7) |
| Agreement: I do not need insurance (mean, sd) | 1.5 | (0.9) | 1.7 | (1.2) |
| Agreement: Insurance is not worth the cost (mean, sd) | 2.3 | (1.3) | 2.4 | (1.4) |
| Agreement: I take more risks than others (mean, sd) | 2.3 | (1.2) | 2.3 | (1.2) |
| Agreement: I can fight illness without a doctor (mean, sd) | 2.5 | (1.3) | 2.4 | (1.2) |
| Agreement: I am healthier than others (mean, sd) | 2.7 | (1.1) | 2.7 | (1.1) |
| Gender Identity (%, n) | | | | |
| Male | 38.1% | 32 | 32.6% | 28 |
| Female | 59.5% | 50 | 67.4% | 58 |
| Non-Binary | 2.4% | 2 | 0.0% | 0 |
| Race (%, n) | | | | |
| White | 91.7% | 77 | 96.5% | 83 |
| Black | 1.2% | 1 | 0.0% | 0 |
| Asian | 2.4% | 2 | 2.3% | 2 |
| AIAN | 3.6% | 3 | 1.2% | 1 |
| Multiple | 1.2% | 1 | 0.0% | 0 |
| Ethnicity (%, n) | | | | |
| Not Hispanic | 92.9% | 78 | 96.5% | 83 |
| Hispanic | 7.1% | 6 | 3.5% | 3 |
| Rurality (%, n) | | | | |
| Urban | 14.3% | 12 | 14.0% | 12 |
| Suburban | 27.4% | 23 | 17.4% | 15 |
| Rural | 58.3% | 49 | 68.6% | 59 |
| Education Level (%, n) | | | | |
| Less than High School | 2.4% | 2 | 1.2% | 1 |
| High school | 27.4% | 23 | 23.3% | 20 |
| Some college, no degree | 25.0% | 21 | 17.4% | 15 |
| Associate's: vocational | 0.0% | 0 | 3.5% | 3 |
| Associate's: academic | 6.0% | 5 | 2.3% | 2 |
| Bachelor's (BA, AB, BS, BBA) | 23.8% | 20 | 32.6% | 28 |
| Master's | 11.9% | 10 | 15.1% | 13 |
| Professional, Doctoral | 3.6% | 3 | 4.7% | 4 |
| Insurance Status (%, n) | | | | |
| Insured - Private | 46.4% | 39 | 45.3% | 39 |
| Insured - Medicare | 21.4% | 18 | 24.4% | 21 |
| Insured - Medicaid | 26.2% | 22 | 23.3% | 20 |
| Uninsured | 6.0% | 5 | 7.0% | 6 |
| Deductible Status (%, n) | | | | |
| Met | 21.4% | 18 | 22.1% | 19 |
| Unmet | 44.0% | 37 | 45.3% | 39 |
| No Deductible | 34.5% | 29 | 32.6% | 28 |
| Health Status (%, n) | | | | |
| No Chronic Condition | 50.0% | 42 | 47.7% | 41 |
| Chronic Condition | 50.0% | 42 | 52.3% | 45 |

## Mixed multinomial logit results

The results of MMNL model are shown in Table 4 where we separated results for X-ray and MRI. We used 1000 iterations in both models. The literature on the number of Halton draws required for valid random parameter estimation with DCE data suggests that, depending on the number of random parameters, stable mixed-logit estimation requires at least 1000 draws. We found that out-of-pocket costs, interpreting doctor specialty, whether or not the clinic was recommended by the primary care provider, the wait time to results, the clinic service rating and online scheduling were all statistically significant and had the expected signs for both MRI and X-ray. Patients were less likely to choose a clinic if the out-of-pocket costs were higher and the wait time to results was longer, but more likely to choose it if images were interpreted by a specialty radiologist, the clinic was recommended to them by their primary care provider, the service rating was higher and if online scheduling was available. For X-ray, free parking was associated with a higher probability of choosing a clinic.

The attributes that mattered the most to patients for both MRI and X-ray were specialty radiologist reading (0.732, standard error = 0.116 for MRI; 0.374, se = 0.122) for X-ray); recommendation by the primary care provider reading (0.626, se = 0.112 for MRI; 0.678, se = 0.155 for X-ray) and the clinic's service rating reading (0.520, se = 0.096 for MRI; 0.451, se = 0.126 for X-ray).

## Latent class analysis

Table 5 shows the results of the Latent Class model where age, gender, education, income, whether someone had more than two previous scans, whether they considered themselves healthier than others, whether they had private insurance and whether they were more likely to take risks than others were the indicator variables estimating the probability of class membership. We compared the results of a 3-class model; 4-class model and 5-class model and found the best model fit for the 2-class model, based on the log likelihood and Bayesian Information Criteria (BIC). We found that 54.7 percent of respondents were in class 1 and 45.3 percent in class 2. We found that females, older patients, whether they had at least two previous scans and those who were less likely to take risks than others were highly predictive of being in class 1 (p < 0.02) which we therefore labeled as the "Experienced Patients" class. Those who had fewer than two previous scans were more likely to be in class 2 (2.7111, p < 0.01) as were male patients (-0.8479, p = 0.02) and younger patients (-1.7197, p = 0.02), which we labeled "New Patients". The results show that someone who had a 1-point increase in the Likert scale for "more likely to take risks than others" was significantly more likely to be in class 2 (-3.1354.7277, p = 0.01).

**Table 4. Results of the MMNL model.**

| | (MRI) | (MRI) | (MRI) | (X-ray) | (X-ray) | (X-ray) |
|---|---|---|---|---|---|---|
| | B | SE | 95% CI | B | SE | 95% CI |
| Out of Pocket Cost | -0.004*** | (0.001) | -0.005,-0.003 | -0.006* | (0.002) | -0.010,-0.001 |
| Interpreting Doctor Specialty | 0.732*** | (0.116) | 0.504,0.960 | 0.374** | (0.122) | 0.135,0.614 |
| Clinic Recommended by PCP | 0.626*** | (0.112) | 0.407,0.845 | 0.678*** | (0.155) | 0.374,0.982 |
| Wait Time to Results | -0.012* | (0.006) | -0.024,-0.001 | -0.013*** | (0.004) | -0.020,-0.006 |
| Travel Time to Clinic | -0.007 | (0.006) | -0.004,-0.018 | -0.018* | (0.008) | 0.001,0.034 |
| Wait Time to Appointment | -0.014 | (0.018) | -0.050,-0.022 | -0.023* | (0.012) | 0.000,0.047 |
| Clinic Service Rating | 0.520*** | (0.096) | 0.332,0.708 | 0.451*** | (0.126) | 0.205,0.698 |
| Free Parking | 0.150 | (0.097) | -0.040,0.339 | 0.669*** | (0.108) | 0.457,0.881 |
| Clinic Parking Lot Distance | -0.041 | (0.023) | -0.087,0.005 | -0.011 | (0.007) | -0.025,0.002 |
| Online Scheduling Availability | 0.288* | (0.112) | 0.068,0.508 | 0.375*** | (0.089) | 0.201,0.550 |

***p < .001,

**p < .01,

*p < .05

**Table 5. Results of the latent class analysis.**

| | Class membership | Standard error |
|---|---|---|
| Male | -.84793** | (.3791) |
| Income (ref: lowest income category) | 0.1477 | (.3979) |
| Age (years) | 1.7196** | (.7559) |
| More than 2 previous scans | 2.7110*** | (.9873) |
| Education (ref: no high school) | -12.9818 | (3.6767) |
| More likely to take risk than others | -3.1353*** | (1.2389) |
| Private insurance | -1.0671 | (1.1982) |
| Healthier than others | -14.4012 | (10.5153 |
| | **Class 1 (Experienced Patients) B (SE)** | **Class 2 (New Users) B (SE)** |
| Costs > current | -0.1774*** (0.0367) | -0.2783 (0. 1764) |
| Specialty read | 0.35123*** (0.0419) | 0.0525 (0. 2753) |
| PCP recommendation | 0.3890*** (0.0432) | 0.1641 (0.2178) |
| Online Scheduling | 0.1240*** (0.0419) | 4.7541*** (0.4703) |
| Service rating | 0.3033***(0.0509) | 0.1148*** (0.2636) |
| Time to Results | -0.0225 (0.0340) | -0.0289 (0.1763) |
| Travel Time | -0.1473*** (0.0324) | -0.2649 (0.1618) |
| Wait Time | -0.1165*** (0.0447) | -0.1271 (0.1968) |
| Access to parking | -0.0308 (0.0254) | -0.0509 (0.1829) |

*** p < .01, ** p < .05, * p < .10

When decentralizing some services away from the main hospital, the focus should be on passing on the medical center's high service rating to the HORD and offering online scheduling. Indeed, patients value easier access, shorter wait times and lower out-of-pocket costs which the growing popularity of RRCs has shown. HORDs can win in popularity by making sure they attain and retain high star ratings while offering better access than RRCs. Primary care providers can potentially play an important role in directing patients to HORDs for their diagnostic imaging services.

For patients in class 1 ("Experienced Patients"), costs (-0.1774), specialty read (0.3512), PCP recommendation (0.3890), travel time (-0.1474), wait time (-0.1165), service (0.3033) and online scheduling (0.1240) all had significant effects (p < 0.01) and had the expected signs. For example, patients in the "Experienced Patients" class will be more likely to choose a clinic location if it is lower cost, takes less travel time, has shorter wait times for results, if results are read by a specialty radiologists, the clinic is recommended by their PCP, service is better and online scheduling is available. Patients in the" New Patients" class only cared about online scheduling (4.7542) and service rating (1.1149), but the effect size was high. None of the other attributes of the service would affect their choice.

## Marginal rates of substitution

Table 6 shows an analysis of the trade-offs patients were willing to make, known as the marginal rates of substitution. We found that for patients in the Experienced Patient class, even though most attributes significantly affected their choice for clinic, the effect size was considerably smaller than for the New Patients class. Experienced Patients were willing to pay: $2 more than what they currently pay (out-of-pocket) to have their images read by a radiologist specialist; $2 more to go to a clinic that was recommended by their PCP; $0.70 more for online scheduling; $1.70 more for a 1-point higher star rating; $0.80 more to have a clinic that would be 1 minutes closer than their current one; $0.65 more for a clinic that had a 1-hour shorter wait and $0.17 more for a clinic that would decrease their walk-up time by 1 minute. The New Patients were willing to pay $17 more for online scheduling and $4 more for a higher star rating.

## Attribute importance

Patients were also asked to rank-order the attributes in how they prioritized them when making a choice for a clinic. This allows us to ask respondents if there was any attribute non-attendance. This means that when processing the attributes, some patients may not consider particular attributes at all. The violin plot in Fig 2 shows the results of the question on attribute importance. The results reflect self-reported order of importance which may not be consistent with how respondents subconsciously assign values to attributes of a choice in the DCE. We see that the results are consistent with the ML results: the most important attributes included interpreting doctor specialty level, PCP recommendation, and costs. There was a small difference between the X-ray and MRI arms: among X-ray patients, service rating and parking access also ranked highly.

**Table 6. Willingness-to-Pay (MRS).**

| WTP for Attribute ($$) | Class 1 (Experienced Patients) | Class 2 (New Patients) |
|---|---|---|
| Radiologist Specialty | 1.980 | 0.189 |
| PCP Recommendation | 2.192 | 0.590 |
| Time to Results | -0.127 | -0.104 |
| Travel Time to Clinic | -0.831 | -0.952 |
| Wait Time to Appointment | -0.657 | -0.457 |
| Service Quality | 1.709 | 4.005 |
| Parking Accessibility | -0.174 | -0.183 |
| Online Scheduling Available | 0.699 | 17.079 |

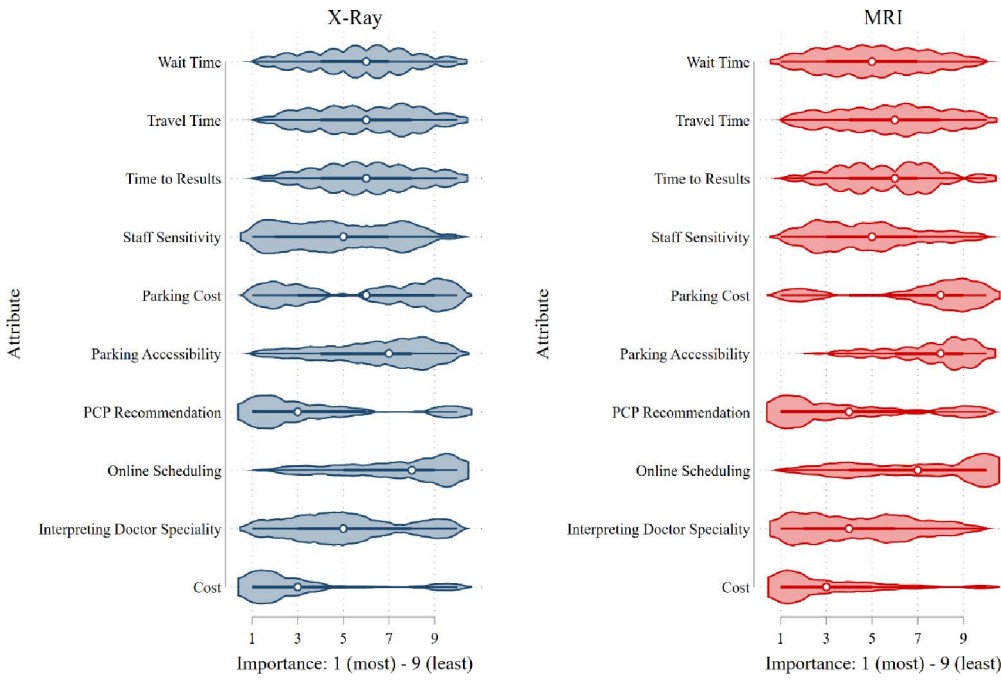

**Fig 2. Attribute importance.**

## Discussion

In this paper, we sought to identify patient preferences for outpatient diagnostic imaging services in the service area of a medium sized academic medical center. We analyzed how patients make trade-offs between attributes of services using a discrete choice experiment. We explored patient's individual preference heterogeneity for these services and reported how preferences vary among and within different groups of patients. In our base analysis, we found that specialty reading of images, PCP recommendation, lower costs, travel time and wait time, as well as higher star rating (representing better service or reputation) and online scheduling are all significant predictors of choice regarding where to get diagnostic imaging services. However, when we segmented the sample population deterministically, we found that males, younger people, and people who are more likely to take risks than others only cared about online scheduling and service rating of the facility. We termed this group "New Patients" as they had significantly fewer previous scans and do not highly value health services. Insurance status, health status and chronic conditions, education and income did not define class membership.

Overall, our analysis showed that the "Experienced Patients" subgroup value diagnostic imaging services in both acute and chronic situations care about different attributes of imaging services in HORDs than "New Patients" who significantly valued the service rating of the hospital outpatient radiology departments (HORD) and online scheduling. This study was performed within the hospital service area of a midsize academic medical center in a rural area in the Northeast of the United States, so it is unclear how these results translate to patient preferences at the national level. External validity remains a challenge for any DCE study even though, while an important component, it has been argued by others that the investigation of external validity should be much broader than a comparison of final outcomes [40].

Overall, however, we conclude from our findings that HORDs can be more competitive by providing services that live up to expectations better than available retail radiology clinics (RRCs). Most RRCs do not currently offer online scheduling, so ease of access may also steer patients towards HORDs. HORDs have the advantage of being linked to the main medical center which has the reputation of having clinical expertise and more sophisticated technology. We conclude that there is room for medical centers to build HORDs that provide an appealing and competitive alternative to the current RRC.

These results also suggest that decision-makers looking to decentralize imaging services while incorporating patient preferences for attributes of those services should differentiate between the different patient sub-populations they are serving. This requires careful consideration of patient characteristics as well as preferences. Overall, we found in this study that New Patients care about the reputation or star rating of a clinic and online scheduling availability. But additionally, Experienced Users – who will be most radiology users – are focused on wait time, price, and recommendations from primary care providers. This may be concrete message to medical centers seeking to decentralize their services away from the main hospital: patients will want to know that the new location offers the same service level in addition to convenience and will be relying on their primary care providers for advice, suggesting outreach to primary care providers will be important for success. Follow-up work, using a larger sample size, should further analyze preference heterogeneity and establish in more detail how trade-offs differ between and within individual patients. Cheaper services may be important to some while the service level matters more to others compared to costs. Studying this preference heterogeneity in more detail will provide a better understanding of these trade-offs and potential take-up of new, decentralized services.

Follow-up work should also establish if there is preference heterogeneity for concierge radiology in general. Concierge services may be decentralized and focus on offering direct access to a subspecialty-trained radiologist, dedicated resources, and a standard turnaround time for image interpretation. A personalized, patient-centered, and attentive approach to image acquisition, interpretation, and reporting leads to a higher level of customer service, but first we need to understand what patient preferences and plan health care service delivery accordingly.

Optimizing patient satisfaction may require a new communication model. This study just focused on these aspects of the services, however, and the authors were not able to assess trade-offs that patients make from the survey data.

## Limitations

Although Centiment used a quota sampling approach, the gender balance may not be reflective of the total population, although we do not believe this is a major threat to external validity. However, while our sample is largely representative of the larger population in this hospital service area of the academic medical center in the Northeast, we measure intention for hypothetical choices and cannot say for sure that these consistently translate to real-life behavioral trade-offs, especially in acute situations. We do expect that the isolated study setting may have influenced the results since respondents in this study do not have very many options in real life. We expect the results to look different for less rural areas where there is already more competition between clinics to offer retail radiology services. Therefore, more work needs to be done to further explore factors that affect decision-making and preferences in these circumstances. We will extend this study to a national DCE including respondents from different areas to see to what extent aspects like rurality, access to care, and choice between different clinics in the region affects attributes of the choice such as parking availability and service level or reputation. We will then also be able to test different heuristics that patients may use when making decisions about seeking care away from their usual sources of care.

The 14 DCE choice sets included 3 alternatives each and 10 attributes with various levels which may be a cognitive burden for respondents. We pre-tested the design with 20 participants and asked them if they felt it was a cognitive burden. None of the respondents answered with "yes". On average, they took 12 minutes to complete the survey.

## Conclusions

In this study, we analyzed the trade-offs patients make between attributes of radiology services to inform decision-making around designing optimal HORDs. Our analysis showed that a patient population can and should be segmented into subgroups that evaluate the value of imaging services differently. The "Experienced Patients" subgroup generally value diagnostic imaging services in both acute and more chronic situations and they had a strong preference for a HORD that would provide services at lower costs, where their images would be interpreted by a specialty radiologist, the clinic would be recommended by their PCP, online scheduling would be available, service rating were higher, and travel and wait times would be shorter. HORDs can therefore be more competitive by providing services that live up to these expectations better than available RRCs. The goal of this study was to get a better understanding of how trade-offs between the attributes are being made and whether these preferences are different for patients who currently frequently visit the hospital for imaging services and those who do not.

What we learned from this study is that most RRCs do not currently offer online scheduling so ease of use may also steer potential future patients towards HORDs. There is an opportunity for hospitals to decentralize some of their service and regain patients to services delivered by the hospital which may help minimize secondary reads. More importantly, HORDs have the advantage of being linked to the main medical center which has the reputation of having clinical expertise and more sophisticated technology. We conclude that there is room for medical centers to build HORDs that provide an appealing and competitive alternative to the current RRC. RRCs have the risk of unreliable quality in technology and imaging interpretation, and it is therefore desirable that HORDs provide the same or more benefits while maintaining the quality care.

## Supporting information

**Appendix 1.** Scenario introduction.
(DOCX)

**Appendix 2.** Demographics participants.
(DOCX)

**Appendix 3.**  Survey text.
(PDF)

## Author contributions

**Conceptualization:** Eline van den Broek-Altenburg, Jamie S. Benson, Kristen K. DeStigter.

**Data curation:** Eline van den Broek-Altenburg, Jamie S. Benson.

**Formal analysis:** Eline van den Broek-Altenburg, Jamie S. Benson.

**Funding acquisition:** Eline van den Broek-Altenburg.

**Methodology:** Eline van den Broek-Altenburg, Adam J. Atherly.

**Project administration:** Eline van den Broek-Altenburg.

**Software:** Eline van den Broek-Altenburg.

**Supervision:** Eline van den Broek-Altenburg, Adam J. Atherly, Kristen K. DeStigter.

**Writing – original draft:** Eline van den Broek-Altenburg.

**Writing – review & editing:** Eline van den Broek-Altenburg, Jamie S. Benson, Adam J. Atherly, Kristen K. DeStigter.

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
