## [Decision Letter · Decision Letter 0]

12 Aug 2024

PONE-D-24-10399Patient Preferences for Diagnostic Imaging Services: Decentralize or not?PLOS ONE

Dear Dr. van den Broek-Altenburg,

Thank you for submitting your manuscript to PLOS ONE. After careful consideration, we feel that it has merit but does not fully meet PLOS ONE’s publication criteria as it currently stands. Therefore, we invite you to submit a revised version of the manuscript that addresses the points raised during the review process.

Both reviewers believe that the paper warrants consideration due to the growing importance of the topic and the research design proposed. However, they also identify numerous limitations that prevent the article from being published in its current form. The two reports are highly insightful and provide detailed guidance for revisions. My recommendation is to follow the suggestions provided on specific points, but also to reconsider the entire  structure of the paper to improve internal consistency and ensure that all relevant elements are adequately addressed in the appropriate section. This is required to achieve the quality standard required by a highly reputed outlet such as Plos One. 

We look forward to receiving your revised manuscript.

Kind regards,

Matteo Lippi Bruni, PhD

Academic Editor

PLOS ONE

Reviewers' comments:

Reviewer's Responses to Questions

**Comments to the Author**

1. Is the manuscript technically sound, and do the data support the conclusions?

Reviewer #1: Partly

Reviewer #2: Partly

2. Has the statistical analysis been performed appropriately and rigorously? 

Reviewer #1: No

Reviewer #2: I Don't Know

3. Have the authors made all data underlying the findings in their manuscript fully available?

Reviewer #1: No

Reviewer #2: No

4. Is the manuscript presented in an intelligible fashion and written in standard English?

Reviewer #1: Yes

Reviewer #2: Yes

5. Review Comments to the Author

Reviewer #1: This is an interesting and important topic given the rising demand. However, the manuscript needs major revisions to be publishable.

Abstract: p2 l44: It is unnecessary to specify the abbreviation again, as HORD has already been defined.

Introduction: p3 l72: The reference needs formatting. This seems to be mistaken.

Introduction: p4 l88: The abbreviation CMS needs to be defined.

Methods: p5 l119-123: This paragraph should be placed in the experimental design section since some of the information given is repeated in that section. Please consider restructuring/rewriting. Nonetheless, the authors need to give more detail on the literature research (where? when? search terms?) as well as the mentioned pilot study. What were the aims? Who was interviewed, etc.? Are there any references on the pilot and literature?

Methods: p6 l144: Until now, the reader has no idea which attributes will be used. The qualitative as well as pilot study have been described without any details. The manuscript clearly lacks results of the qualitative study parts.

Experimental design: How many levels per attribute were used? The manuscript would benefit from a table showing attributes and levels. What about the actual design? Was it a full design? Which software was used to create the design?

Data source: p7 l189: Which consistency checks were used? What was the bot-behavior check, and how did the authors check for failing attention? This needs to be clarified.

Statistical Methods & Latent Class Analysis: Why are these separate paragraphs? The information given is quite repetitive. Please be more precise.

Table 2: p11: Please define your attributes. Without an understanding of how the attributes (and levels?) were defined and described for the participants, it is not possible to conclude anything from this table. Crucial information is missing, which is a major deficit of the manuscript.

Latent Class Analysis: p12: How many analyses were performed? Did you just perform the two-class model or 3-, 4-, and 5-class models as well? What was the rationale for choosing the two classes? It seems too easy to just present the two-class option without discussing other options or at least reasons for rejecting different solutions.

Attribute importance: p15: Do you have any ideas on why the results of the rating are not completely equal to the DCE?

Discussion: p16: The discussion is rather short and superficial. To what extent may the university setting have influenced the results? Might the results look different for a rural area with no university medical center nearby?

Limitations: p16: I think a major limitation is the inclusion of only one university medical center. Due to that, the representativeness of the results must be questioned.

General: The manuscript has too many repetitions and should be rewritten/restructured to be more precise.

Reviewer #2: The aim of this paper is to employ a discrete choice experiment (DCE) to determine individuals' preferences for outpatient diagnostic imaging services. This inquiry is particularly pertinent given that the rising demand for such technologies may necessitate increased availability, and the way to move forward must depend on people’s preferences. The chosen methodology is appropriate, as a DCE is effective in ranking preferences for various attributes and levels, as well as in calculating the willingness to pay for those specific attributes. Additionally, techniques like latent class analysis within a DCE can be used to explore how preferences vary across different demographic groups, which can aid in the more tailored design of services to better meet the needs of the populations they are intended to serve.

Although the paper generally employs sound methodologies, there are several critical issues that need to be addressed. The most significant of these concerns is the insufficient explanation of the methodology and design, which is particularly vital in a DCE. Additionally, the paper lacks a cohesive narrative and clear policy implications. Below, I will outline my comments in the order that these issues appear in the document:

Abstract:

Please, do not use acronyms in the abstract.

Introduction:

Line 72, page 3: “(…) back on the same patient within a 3-week period 5.” – Is the 5 a typo?

Lines 78-80, page 3: “Some hospital systems are now looking at options to develop their own RRCs to provide convenient and efficient locations, without sacrificing the technological or interpretation quality, while leveraging the existing trust patients may have with their hospital system” – Any reference to support this claim?

Lines 80-82, page 3: “This is a potentially valuable innovation for patients, as they would be able to receive the “specialty read” quality at locations with shorter wait times and lower out-of-pocket costs.” – In my opinion this is a problem regarding the narrative of this paper. What the paper ends up saying is that patients want the traditional service (which is superior), but faster and cheaper. I don’t think this is enough as a policy implication for this paper, since there is no need for a DCE to know this.

Lines 91-96, page 4: “Existing research on patient preferences in diagnostic imaging services is sparse, however, and what exists is often contradictory 9-15. Though survey studies have explored from whom patients prefer to learn their results and how (e.g., by phone, e-mail, or in-person), findings to date have been inconsistent 9–11,16. For example, one study found a majority preference for learning results in-person and from a radiologist10 while another found that most patients prefer to learn results from their primary care provider, over the phone.” – Is this all? This seems like a very limited discussion of the literature. Can you please provide more depth to this description of the literature?

Methods:

Line 152, page 6: “Situation 1): ”For the purpose of this study, suppose you hurt your arm (…)” – Is there a reason why it is an “arm” for the x-ray and “back” fort the MRI.

Line 153, page 6: “You have three options (…)” – Is there a reason why you chose 3 alternatives instead of the traditional 2 alternatives in a DCE? Is there any concern that 3 alternatives and 14 choice sets may be too much? Any discussion on potential cognitive burden?

Line 162, page 6: “These selected attributes included (…)” – Where are the levels? It is impossible to review a DCE paper without the levels. Can you please provide a table with all attributes and respective levels? You also need a lot more detail on how the attributes and (especially) levels were chosen. Also, the example choice set in appendix lacks resolution.

Line 182, page 7: “Centiment contacted approximately 472 (…)” – Is it approximately 472 or exactly 472?

Line 188-190, page 7: “We excluded 98 subjects for failing consistency criteria, qualifying for a closed quota, failing a bot-behavior check, or failing an attention (response quality) check” – Can you provide more detail? What were these consistency criteria or attention checks, for example?

Line 190, page 7: “final sample of 170: 84 in Xray, 86 in MRI.” – Are these enough? I know you have 14 choice sets with 3 alternatives each, but still an unusually low number of observations since you are dividing the analysis into two DCE, with later latent class analysis. Care to comment on what lead you to believe this is enough? You say in line 93, page 8 that “We used NGene 1.2.1 (ChoiceMetrics, 192 2018) to estimate the minimum sample size required for this study.”, but what methodology is this?

Line 205-211, page 8 – I would cut this entire paragraph. I don’t think it adds anything to the paper.

Results:

Line 222-223, page 9: “receiving the X-ray choice questions had had an average of 4.8 previous images while MRI patients had had 2.9 .” Is this per year?

Line 228, page 10: “Mixed logit Results” – How many iterations did you use in the mixed logit?”

Line 249, page 12: “Latent Class Analysis” – The latent class analysis is for which DCE? The Xray or MRI?

Line 254, page 12: “We found that 54.7 percent of patients were in class 1 and 45.3 percent in class 2” – Why 2 classes? Which criteria did you use?

Line 256, page 12: “of being in class 2 (p<0.02) which we therefore labeled as the “Experienced Patients” class” – I think there is a mistake here because in the table the experienced patients are class 1 and not 2. Also, why did you choose the terms “Experienced Patients” and “New Patients”. Not sure the text explains that choice very well.

Line 269 to 275, page 13 – I think the font is different in this section

Line 296-297, page 14 – “$2 more to go to a clinic that was recommended by their PCP; $0.70

more for online scheduling; $1.70 more for a 1-point higher star rating; $0.80 more to have a

clinic that would be 1 minutes closer than their current one; $0.65 more for a clinic that had a

1-hour shorter wait and $0.17 more for a clinic that would decrease their walk-up time by 1

minute. The New Patients were willing to pay $17 more for online scheduling and $4 more for

a higher star rating.” – I find it hard to evaluate the willingness to pay part of the paper without the levels of the out of pocket in the DCE.

Discussion:

Line 327, page 15: “others only We termed this group” – typo?

Line 337-341, page 16: “This is a concrete message to medical centers seeking to decentralize their services away from the main hospital: patients will want to know that the new location offers the same service level in addition to convenience and will be relying on their primary care providers for advice, suggesting outreach to primary care providers will be important for success.” – This is basically saying that decentralized services must keep the characteristics of the traditional services. But I am not sure this is possible. Therefore, this does not seem like a viable narrative/policy implication. If it was possible to create a better, faster, cheaper, more convenient service than it would have been created. The issue here is that tradeoffs need to be made. So, your question should be, for example, how much cheaper certain services would have to be to compensate for some lack of quality.

Limitations:

Line 344-349, page 16 - There is so much more to say about limitations. The authors never comment on the sample size or the complexity of the design (14 choice sets with 10 attributes and 3 alternatives is a lot). Certainly, there is something to say about the attributes and levels and how they were chosen, but no details are offered in the text. The authors never show the survey (which should be in appendix). There must be something in the survey and how people may have interpreted things that should be noted in the limitations.

Conclusion:

Line 356, page 16: “(…) diagnostic imaging services in both acute and more chronic situations (…)” – Is there anything in the paper about acute vs. chronic?

Line 356-360, page 16-17: “had a strong preference for a HORD that would provide services at lower costs, where their images would be interpreted by a specialty radiologist, the clinic would be recommended by their PCP, online scheduling would be available, service rating were higher, and travel 359 and wait times would be shorter.” – This is just saying they want it all. Who doesn’t? This goes back to my earlier comment that your conclusions end up being that people just want more (and better) of everything. But that is not a new finding. You need to rearrange the narrative of the paper to talk more about trade-offs and how the information you provide on trade-offs can help policy making.

Thank you for the opportunity to read your paper and thank you for your effort!

6. PLOS authors have the option to publish the peer review history of their article (what does this mean? ). If published, this will include your full peer review and any attached files.

**Do you want your identity to be public for this peer review?** For information about this choice, including consent withdrawal, please see our Privacy Policy .

Reviewer #1: No

Reviewer #2: **Yes: ** Luis Filipe

---

## [Author Response · Author response to Decision Letter 1]

28 Sep 2024

Editor's comments

https://journals.plos.org/plosone/s/file?id=wjVg/PLOSOne_formatting_sample_main_body.pdf andhttps://journals.plos.org/plosone/s/file?id=ba62/PLOSOne_formatting_sample_title_authors_affiliations.pdf

We edited the manuscript and title page according to the PLOS ONE style templates.

We added to the methods and online submission that consent was informed through a written statement at the start of the online survey. Without the online consent, respondents were not able to proceed. The answers were recorded in the data.

We have noted this in the Methods section of the manuscript.

3. . We note that the grant information you provided in the ‘Funding Information’ and ‘Financial Disclosure’ sections do not match.

We matched the funding information with financial disclosure. We added the appropriate awards number.

We provided a completed Data Availability Statement in the submission form. We have now shared our data from this study to a public repository as well as with a supporting information file with this submission. The public repository can be found here: https://github.com/ElinevandenBroek/DecentralizedImaging

We do include a separate caption for each figure.

Response to Reviewers

Reviewer #1: This is an interesting and important topic given the rising demand. However, the manuscript needs major revisions to be publishable.

We very much appreciate your feedback and respond to each point raised below.

Abstract: p2 l44: It is unnecessary to specify the abbreviation again, as HORD has already been defined.

Thank you, we deleted second mention at p2 l44.

Introduction: p3 l72: The reference needs formatting. This seems to be mistaken.

Thank you, we formatted the reference.

Introduction: p4 l88: The abbreviation CMS needs to be defined.

We added Centers for Medicare and Medicaid Services.

Methods: p5 l119-123: This paragraph should be placed in the experimental design section since some of the information given is repeated in that section. Please consider restructuring/rewriting. Nonetheless, the authors need to give more detail on the literature research (where? when? search terms?) as well as the mentioned pilot study. What were the aims? Who was interviewed, etc.? Are there any references on the pilot and literature?

We completely rewrote the methods section to include a lot more detail about the experimental design, the aim of the study, the participants and the pilot data. We also moved the results of the qualitative study to the results section.

Methods: p6 l144: Until now, the reader has no idea which attributes will be used. The qualitative as well as pilot study have been described without any details. The manuscript clearly lacks results of the qualitative study parts.

In the new text, the paragraph in the Methods section now reads as:” We conducted focus groups among patients at a middle-size academic medical center who volunteered to share their experiences with imaging services. The results of that research are reported in the results section. Following the analysis of the focus groups, we included the following attributes: whether the interpreting radiologist is a general or sub-specialty radiologist; whether the clinic was recommended by their primary care physician (PCP); time to results; out of pocket cost; wait time to an appointment; travel time to the clinic; parking costs; parking accessibility; service; and whether or not online scheduling is available. Service is a multifactorial attribute (e.g., staff attentiveness and facility amenities) combined into a star rating. The rating scale is between one and five stars, with a five-star rating representing an excellent service and a one-star rating suggesting a poor service, as rated by other hypothetical patients.”

We deliberately moved the results to the Results section, including the results of the focus groups. It is now clear in the Methods section how the DCE was constructed and the Results section clarifies the specific findings.

Experimental design: How many levels per attribute were used? The manuscript would benefit from a table showing attributes and levels. What about the actual design? Was it a full design? Which software was used to create the design?

The text now also clarifies: “Each choice task had three different clinics which varied by 10 different attributes, which were determined by prior qualitative research. Four attributes had 2 levels and six attributes had 3 levels.”

We clarified in the text that we used the software NGene to design an experimental design. We also included a table with all the attributes and levels.

Data source: p7 l189: Which consistency checks were used? What was the bot-behavior check, and how did the authors check for failing attention? This needs to be clarified.

Bots were identified by the RegEx program and manual review of the free text entry boxes. Typically, the bots in our sample entered nonsense or repeated the question’s text into those fields. We deleted respondents with w=either missing,"1","6","dsfvrg rfg", "mroe ogod", "the best", "very easy" in their answers.

For attention, we then filtered by completion time over a threshold and also filtered any respondents out that showed straight-lining behavior.

We added this clarification to the text.

Statistical Methods & Latent Class Analysis: Why are these separate paragraphs? The information given is quite repetitive. Please be more precise.

For our analytic approach, we used mixed logit models and latent class analysis. LCA was a sub-paragraph of statistical methods. We took out the subtitle and checked for repetition.

Table 2: p11: Please define your attributes. Without an understanding of how the attributes (and levels?) were defined and described for the participants, it is not possible to conclude anything from this table. Crucial information is missing, which is a major deficit of the manuscript.

We added a Table in the Methods section with all the attributes and levels to provide an understanding of how the attributes and levels were defined.

Latent Class Analysis: p12: How many analyses were performed? Did you just perform the two-class model or 3-, 4-, and 5-class models as well? What was the rationale for choosing the two classes? It seems too easy to just present the two-class option without discussing other options or at least reasons for rejecting different solutions.

We did compare the results of a 3-class model; 4-class model and 5-class model and found the best model fit for the 2-class model, based on the log likelihood and Bayesian Information Criteria (BIC). We explained this in the text.

Attribute importance: p15: Do you have any ideas on why the results of the rating are not completely equal to the DCE?

These are not equal to the DCE since in our models, we estimate relative utilities. We added text to this paragraph to clarify.

Discussion: p16: The discussion is rather short and superficial. To what extent may the university setting have influenced the results? Might the results look different for a rural area with no university medical center nearby?

We added text to the discussion section to address these questions.

Limitations: p16: I think a major limitation is the inclusion of only one university medical center. Due to that, the representativeness of the results must be questioned.

We agree with the reviewer that we should be conservative with regard to out-of-sample predictions. We therefore added some text to the limitations sections and also checked the Discussion section for consistency.

General: The manuscript has too many repetitions and should be rewritten/restructured to be more precise.

We have completely rewritten the manuscript and restructured some sections.

Reviewer #2: The aim of this paper is to employ a discrete choice experiment (DCE) to determine individuals' preferences for outpatient diagnostic imaging services. This inquiry is particularly pertinent given that the rising demand for such technologies may necessitate increased availability, and the way to move forward must depend on people’s preferences. The chosen methodology is appropriate, as a DCE is effective in ranking preferences for various attributes and levels, as well as in calculating the willingness to pay for those specific attributes. Additionally, techniques like latent class analysis within a DCE can be used to explore how preferences vary across different demographic groups, which can aid in the more tailored design of services to better meet the needs of the populations they are intended to serve.

Although the paper generally employs sound methodologies, there are several critical issues that need to be addressed. The most significant of these concerns is the insufficient explanation of the methodology and design, which is particularly vital in a DCE. Additionally, the paper lacks a cohesive narrative and clear policy implications. Below, I will outline my comments in the order that these issues appear in the document:

Abstract:

Please, do not use acronyms in the abstract.

We have specified all acronyms with full names but since we do use them repeatedly throughout the abstract.

Introduction:

Line 72, page 3: “(…) back on the same patient within a 3-week period 5.” – Is the 5 a typo?

Thank you, yes, this was supposed to be a subscript to refer to the reference which we have now fixed.

Lines 78-80, page 3: “Some hospital systems are now looking at options to develop their own RRCs to provide convenient and efficient locations, without sacrificing the technological or interpretation quality, while leveraging the existing trust patients may have with their hospital system” – Any reference to support this claim?

We added the relevant references to the claim.

Lines 80-82, page 3: “This is a potentially valuable innovation for patients, as they would be able to receive the “specialty read” quality at locations with shorter wait times and lower out-of-pocket costs.” – In my opinion this is a problem regarding the narrative of this paper. What the paper ends up saying is that patients want the traditional service (which is superior), but faster and cheaper. I don’t think this is enough as a policy implication for this paper, since there is no need for a DCE to know this.

The background of this study is that a midsize academic medical center in the Northeast considered building a HORD and decentralizing some services away from the hospital. Partly this was due to commercial competition with pop-up imaging services and partly because of the challenge the pop-ups bring for necessary secondary reads in the tertiary care center. From our qualitative work, we know that many patients do want the traditional service but end up going to pop-up services (for example, from the back of a car or truck) because it is cheaper and more accessible. So, when organizing a decentralized service from the hospital, the main question was not so much what attributes of the service patients care about, but the tradeoffs they make when they decide where to get these services they need. The marginal rates of substitution are important to inform the business model of a new to establish decentralized clinic. This is why we decided to do a DCE and elicit patient preferences and study preference heterogeneity. The hope was to predict uptake based on different levels of attributes.

Lines 91-96, page 4: “Existing research on patient preferences in diagnostic imaging services is sparse, however, and what exists is often contradictory 9-15. Though survey studies have explored from whom patients prefer to learn their results and how (e.g., by phone, e-mail, or in-person), findings to date have been inconsistent 9–11,16. For example, one study found a majority preference for learning results in-person and from a radiologist10 while another found that most patients prefer to learn results from their primary care provider, over the phone.” – Is this all? This seems like a very limited discussion of the literature. Can you please provide more depth to this description of the literature?

We have added text to substantiate the various references in this paragraph ads we cite 7 studies. It now reads:

Existing research on patient preferences in diagnostic imaging services is sparse, however, and what exists is often contradictory 9-15. One study looked at patient preferences for CT and MRI imaging results and focused on how, from whom, and how soon patients preferred the results 10. The authors used a survey and asked those questions directly of patients. They found that patients wanted their results communicated much sooner than is currently practiced. Optimizing patient satisfaction may require a new communication model. This study just focused on these aspects of the services, however, and the authors were not able to assess trade-offs that patients make from the survey data. Another study focused receiving radiology results and reviewing the images and findings directly with a radiologist after completion of an examination 11. Patients prefer hearing examination results from both their ordering provider and the interpreting radiologist. These results were consistent with another study focused on results communication 12. These are interesting findings, however, only focus on results and not preferences for where to seek services. Another study focused on the referral mechanism 13 and the importance of wait time for test results 14. None of these studies focused on describing which aspects of imaging services were important and preferred by patients, which has bene done in other fields such as preferences for primary care services 15 and preferences for health service delivery in general 16.

Methods:

Line 152, page 6: “Situation 1): ”For the purpose of this study, suppose you hurt your arm (…)” – Is there a reason why it is an “arm” for the x-ray and “back” fort the MRI.

We looked at most common reasons for acute X-ray which included fractures. Arm fractures are the most common. For MRI, one of the most common uses is with spinal injuries or low back pain.

We added “based on common reasons for imaging services” to the manuscript text.

Line 153, page 6: “You have three options (…)” – Is there a reason why you chose 3 alternatives instead of the traditional 2 alternatives in a DCE? Is there any concern that 3 alternatives and 14 choice sets may be too much? Any discussion on potential cognitive burden?

There is not a “traditional” number of alternatives in a DCE. We create an efficient design described in the methods section where we need a combination of minimum sample size,

---

## [Decision Letter · Decision Letter 1]

13 Nov 2024

PONE-D-24-10399R1

Patient Preferences for Diagnostic Imaging Services: Decentralize or not?

PLOS ONE

Dear Dr. van den Broek-Altenburg,

Thank you for submitting your manuscript to PLOS ONE. After careful consideration, we feel that it has merit but does not fully meet PLOS ONE’s publication criteria as it currently stands. Therefore, we invite you to submit a revised version of the manuscript that addresses the points raised during the review process.

The two referees have now completed their evaluation of your revised manuscript. They have apprecaited your effort in addressing the feedback provided in the initial review, and both acknowledge that the manuscript has shown improvements. However, several remaining issues still require your careful attention . The  comments included in both reports are comprehensive, and I strongly encourage you to address each of them with great care because a thorough revision is still required to meet the standards for publication.

We look forward to receiving your revised manuscript.

Kind regards,

Matteo Lippi Bruni, PhD

Academic Editor

PLOS ONE

Reviewers' comments:

Reviewer's Responses to Questions

**Comments to the Author**

1. If the authors have adequately addressed your comments raised in a previous round of review and you feel that this manuscript is now acceptable for publication, you may indicate that here to bypass the “Comments to the Author” section, enter your conflict of interest statement in the “Confidential to Editor” section, and submit your "Accept" recommendation.

Reviewer #1: (No Response)

Reviewer #2: (No Response)

2. Is the manuscript technically sound, and do the data support the conclusions?

Reviewer #1: No

Reviewer #2: Yes

3. Has the statistical analysis been performed appropriately and rigorously? 

Reviewer #1: No

Reviewer #2: Yes

4. Have the authors made all data underlying the findings in their manuscript fully available?

Reviewer #1: No

Reviewer #2: No

5. Is the manuscript presented in an intelligible fashion and written in standard English?

Reviewer #1: Yes

Reviewer #2: Yes

6. Review Comments to the Author

Reviewer #1: During the revision, the manuscript has improved. The issues raised have been addressed by the authors. However, the manuscript still needs revisions to be publishable. Especially the methods section needs clarification.

Introduction: p4 l.91-104: This section is not an introduction to me, but discussion. I suggest moving the paragraph to the discussion section.

Methods - general: The whole section needs more references! Authors need to declare the references used. There are numerous pages without any reference in this section. Moreover, the methods section needs restructuring and/or subheadings. Right now, it is confusing since qualitative study parts and the description of the experimental DCE design mix up.

Methods: p5 l.117-124: This paragraph is not related to experimental design of the DCE. It is rather an explanation of the underlying problem of preference elicitation in all-day routine. Maybe move to the introduction.

Methods: p6 l.138-143: This is not related to experimental design of the DCE, but a description of the qualitative study. Please clarify. Nonetheless, the authors need to give more detail on the literature research (where? when? search terms?) as well as the mentioned qualitative research. What were the aims? How were potential attributes defined? Are there any references?

Methods: p6 l144: Until now, the reader has no idea how attributes were defined. Instead, authors just mention “Four attributes had 2 levels and six attributes had 3 levels “. The qualitative as well as pilot study have been described without any details. The manuscript clearly lacks results of the qualitative study parts. Moreover, I would suggest adding Table 1 directly to the statement on the final number of attributes/levels.

Methods: p6 l145-156: This is not related to experimental design of the DCE, but attribute selection. It should be placed upfront.

Experimental design: The manuscript would benefit from more detail on the actual design. Any overlaps used? What about consistency tests?

Methods: p7/8 l190-205: Suggest adding a subheading since these paragraphs deal with survey/questionnaire design rather that experimental design of the DCE

Data source: p10 l211ff: What is the Centiment panel? Is it open to anybody or linked to a certain institution? More information on the panel would be nice.

Data source: p10 l211ff: Which consistency checks were used? What was the bot-behavior check, and how did the authors check for failing attention? This still needs to be clarified.

Data source: p10 l221: Which quota was used?

Data source: p11 l232: Authors state that the minimum sample size was estimated using NGene. What was the result?

Statistical Methods: This section lacks references!

Qualitative research: p11 l232: As mentioned earlier, some paragraphs from the introduction can be added here

Qualitative research: p12 l259: “From the thematic analysis of the focus groups” – Which qualitative method (and/or software) was used? Did authors rely on any theoretical frame? In addition, this section is lacking real results. How many participants? What was their background and how were they recruited?

Mixed Logit results:p14: Why was the number of iterations set to 100? Any rationale on that?

Latent Class Analysis: Thank you. My comments from the first review have all been answered. Well done.

Limitations: p16: I think a major limitation is the inclusion of only one university medical center. Due to that, the representativeness of the results must be questioned.

General: The manuscript should be rewritten/restructured to be more precise, especially in the methods section.

Reviewer #2: I would like to thank the authors for their corrections.

However, I still have a few concerns I would like to see addressed.

Note that the pages and lines I refer to are the ones in the track changes version.

Page 4, line 113 – “which has bene done in other fields” – typo, correct to “been”.

Page 5, line 132 – “One option to measure hard to measure choice attributes is to use stated preferences (SP) data” – please avoid using more than two or three acronyms throughout the paper. In this case, you use SP so rarely that it probably didn’t even save more than two or three words.

Page 8, line 203 to 205 – “In some cases, a study may contain more choice tasks than the researcher wants to ask respondents. In this case, a blocking experimental design can be used.” – I don’t think this sentence is very well written, to the point that it took me several readings to not understand the opposite of what you want to say. The case for using blocks is usually related to the need for a very big design and fear of cognitive burden. In those cases, researchers may split the design into 2 or more blocks so that each person answers a fewer amount of choice sets, avoiding cognitive burden and tiredness and therefore assuring the quality of the answers. I assume you want to say something within these lines, but that was not what was coming through.

Page 8, line 208 to 210 – “In our study, respondents were asked 6 questions regarding X-ray services and 6 questions regarding Magnetic Resonance Imaging (MRI) services.” – Here I am confused. First, are these two block of the same design or two different designs? While you are using the same design for both, preferences may differ for each service, therefore, I would think you would have to randomly run the full design separately, which would count as two designs, even if they are equal. Second, is it correct to run the same design for both (e.g., shouldn’t some level values be different depending on the service?). Third, 6+6 is 12, but you say they had 14 choice sets. Where are the other two? Also, page 13, line 294 “134 were assigned to the arm X-ray group, 134 to the back MRI group.” Seems to imply they are just two different groups. I apologise if I completely misinterpreted what you wrote, but as of now, I think this part of the manuscript is not very clear.

Page 9 – line 226 – Thank you for adding this table. Since the table in the choice sets needs to be simpler for visual simplicity reasons, I would like to know if respondents received instructions on the attributes and levels. This is way I stress that you should include the survey (or at least important parts of the survey) in appendix, for matters of clarity and for the reviewers to be able to properly evaluate the merits and limitations of the paper. Also, within table one, in the parking attribute, do you specify how much people need to pay in the level where they pay? Also, wouldn’t this be extremely correlated with the costs attribute? I see the point of the parking accessibility, since this implies other levels of stress in trying to park close to the place, but the cost is a little bit harder to understand. What is done is done, but it might deserve a comment from you. Another issue I detected is the Service attribute where I don’t know what these stars mean. Is it one star out of 5? What do they mean? Did respondents have access to this information (or maybe this is obvious to anyone living in the US?).

Page 13, line 299 to 302 – “Bots were identified by the RegEx program and manual review of the free text entry boxes. Typically, the bots in our sample entered nonsense or repeated the question’s text into those fields. We deleted respondents with w=either missing,"1","6","dsfvrg rfg", "mroe ogod", "the best", "very easy" in their answers.” - Sorry, I didn't want you to add this. It was just to be sure what type of attention checks you might have used. For example, some people have included repeated choice sets to try to identify inconsistencies. Some people check the time taken to answer (which you have now included). I was just confirming if you had included any attention check in the survey itself that deserved being referred.

Page 15, line 339 to 342 – “A question guide aimed at understanding how patients perceive their radiology experiences led participants through their focus group session. Questions were intended to be exploratory and were sometimes followed by probes to allow differences between patient insights and experiences to emerge.” – Could you add relevant material related to the focus groups in appendix? For example, you state that you had a list of questions. Could you add those? Also, could you add some description of the participants? Do you think they were somewhat representative. I know you won’t have fully representative participants in a focus group, but were you able to have some diversity (gender, age, ethnicity, social background, etc…)

Page 15, line 352 – “Figure 1 shows an example of a choice task in the DCE” – Is this appendix or is this supposed to be in the main paper? If it is appendix, refer it is in appendix. If not, you don’t need to answer this comment. Just a note that the picture is still blurry and should be corrected at some point.

Page 17, line 368 – “Mixed logit results” – I find it strange you present no comments on heterogeneity in this section? Mixed logit models are one of the two major alternatives of dealing with heterogeneity. So, I would expect you to report and comment on heterogeneity in this section. Also, why did you decide to go with both a mixed logit and a latent class model? Were you unsure if unobserved preference followed a continuous or discrete distribution and wanted to test both? If that is the case you should have said that and comment on both results.

Page 23, line 482 – “relatively rural area in the Northeast” – always write Northeast of the Unite States to clarify.

Page 23, line 485 – “Most RRCs do not currently offer online scheduling so ease of use may also steer patients towards HORDs.” – I think a comma is needed for the sentence to be easier to read. Also, the “ease of use” expression bugs me a little. I understand what you want to say, but the idea of the RRCs was also to increase access, correct? Maybe it is just me, but I tend to put ease of access in the mix for ease of use. If you agree with, maybe consider changing or clarifying the expression.

Page 23, line 501 to 502 “and what would - “push them over the fence” to seek services elsewhere” – Just highlight that your paper is already providing important information for that type of analysis. Knowing how much people value certain characteristics will inform if services can have enough of those to make people using them, and if they can make it a cost that is inferior to peoples willingness to pay, so they can increase the prices to cover those costs while capturing more demand.

Page 24, line 520 to 522 – “Moreover, based on our qualitative research we used a complex design for the DCE, although NGene showed that we had sufficient sample size for both X-ray and MRI.” – This sentence seems to have parachuted itself into the middle of another paragraph. Please, find another place for it. Also, it still does not comment on one possible limitation of your paper, which is whether it is likely that people suffered from any cognitive burden in your analysis, since 3 choice sets with 10 attributes and 14 choices sets is an extremely complex design.

Page 25, line 546 to 549 – “The marginal rates of substitution in terms of costs were shown in table 5; followup work will focus on other trade-offs such as how much travel time patients are willing to trade-off for service or shorter wait time to results. In a national follow-up study, we will collect more data to this analysis.” – You should not refer to a table in the conclusion. You also should not say that some more work is going to be done. Instead, you may say that through your study, further research was identified as needed (but only if this is the case). In general, I don’t like this entire sentence and don’t understand its purpose.

Thank you again for your revision!

7. PLOS authors have the option to publish the peer review history of their article (what does this mean? ). If published, this will include your full peer review and any attached files.

**Do you want your identity to be public for this peer review?** For information about this choice, including consent withdrawal, please see our Privacy Policy .

Reviewer #1: No

Reviewer #2: **Yes: ** Luis Filipe

---

## [Author Response · Author response to Decision Letter 2]

7 Jan 2025

Response to Reviewers

Reviewer #1: During the revision, the manuscript has improved. The issues raised have been addressed by the authors. However, the manuscript still needs revisions to be publishable. Especially the methods section needs clarification.

Thank you, we very much appreciate your additional comments to improve the manuscript. We respond to each point raised below.

Introduction: p4 l.91-104: This section is not an introduction to me, but discussion. I suggest moving the paragraph to the discussion section.

The intention of this paragraph was to introduce the gap in the literature. In response to this comment the paragraph was rewritten and some of the material was moved to the discussion. We have deleted the sentence “To successfully decentralize imaging services at a larger scale, however, it is important to define value not just from the perspective of the healthcare system, but also from the perspective of the patient.” In the previous paragraph to make clear this is purely a description of the literature.

Methods - general: The whole section needs more references! Authors need to declare the references used. There are numerous pages without any reference in this section. Moreover, the methods section needs restructuring and/or subheadings. Right now, it is confusing since qualitative study parts and the description of the experimental DCE design mix up.

The Methods section largely describes the methods we used for this particular study. We added several references to clarify the analytic approach or refer to others explaining the emthods; the remainder is a description of the current study.

We completely reorganized the structuring of the methods section as well to separate the finding of the qualitative work which informed the DCE design. The methods section now starts with the qualitative research. We hope the new structure takes away any confusion.

Methods: p5 l.117-124: This paragraph is not related to experimental design of the DCE. It is rather an explanation of the underlying problem of preference elicitation in all-day routine. Maybe move to the introduction.

We deleted the first few paragraphs of the Methods section and cite this work that describes stated preferences design in detail.

Methods: p6 l.138-143: This is not related to experimental design of the DCE, but a description of the qualitative study. Please clarify. Nonetheless, the authors need to give more detail on the literature research (where? when? search terms?) as well as the mentioned qualitative research. What were the aims? How were potential attributes defined? Are there any references?

We very much appreciate the comments from the reviewer, as we realized a description of this work was missing in prior versions. We added a detailed description of the qualitative work which the Methods section now starts with.

Methods: p6 l144: Until now, the reader has no idea how attributes were defined. Instead, authors just mention “Four attributes had 2 levels and six attributes had 3 levels “. The qualitative as well as pilot study have been described without any details. The manuscript clearly lacks results of the qualitative study parts. Moreover, I would suggest adding Table 1 directly to the statement on the final number of attributes/levels.

Methods: p6 l145-156: This is not related to experimental design of the DCE, but attribute selection. It should be placed upfront.

After restructuring the methods section and adding detailed information about the qualitative research, the construction of the levels should now be clarified. We appreciate the suggestion to add Table 1 (now Table 2) in the Methods section, so we moved it up.

Experimental design: The manuscript would benefit from more detail on the actual design. Any overlaps used? What about consistency tests?

This paper is written for a broad audience and we tried to not be too specific, although we already included a lot of detail on the design. Per reviewers’ request we added text to the experimental design section including regarding overlaps. Consistency tests are described in the data section.

Methods: p7/8 l190-205: Suggest adding a subheading since these paragraphs deal with survey/questionnaire design rather that experimental design of the DCE

Thank you for the suggestion. We added another subheading here, separating the DCE from the experimental design description. We also added Figure 1 illustrating a choice question.

Data source: p10 l211ff: What is the Centiment panel? Is it open to anybody or linked to a certain institution? More information on the panel would be nice.

We have added text to this section:

“Centiment is a survey company which recruits individuals to answer surveys to generate rewards for themselves or to pledge their earnings to a nonprofit of their choice and it is open to anyone to participate. Centiment has engineered complex systems to manage their respondents and ensure they are providing thoughtful responses.”

Data source: p10 l211ff: Which consistency checks were used? What was the bot-behavior check, and how did the authors check for failing attention? This still needs to be clarified.

Thank you for this comment. Responding to the previous comment about this, we added a clarification to the text:

“Bots were identified by the RegEx program and manual review of the free text entry boxes. Typically, the bots in our sample entered nonsense or repeated the question’s text into those fields. We deleted respondents with w=either missing,"1","6","dsfvrg rfg", "mroe ogod", "the best", "very easy" in their answers.”

We have now done three additional things to clarify:

- We have made the threshold more explicit: “For attention, we then checked for consistency and filtered by completion time over a threshold of 7 minutes” since it is not possible to read and respond to the questions within that period of time.

- We have clarified what straight-lining means: and also filtered any respondents out that showed straight-lining behavior, meaning that a respondent would always pick the same response to the choice questions”. We added three references here.

- We added to the text regarding bot-behavior and clarified which respondents were deleted for suspected bot behavior. We also added: “In addition, we seeked to identify fraudulent data by defining a priori indicators that warranted elimination or suspicion; an approach borrowed from another study 26.” We added a reference to a relevant study here as well to clarify the approach.

Data source: p10 l221: Which quota was used?

The quotas that were agreed on included 5 percent of census on age by decade, gender, and region; 3 percent of census on race/ethnicity; and 50 percent above and below national median income of $65k/year. We added this clarification to the text.

Data source: p11 l232: Authors state that the minimum sample size was estimated using NGene. What was the result?

We added a clarification to the text: “The S-efficient design we generated in NGene showed that we needed a minimum of 55 respondents so our sample size was this was a sufficient”.

Statistical Methods: This section lacks references!

We added several relevant references, including:

30. Hensher DA, Greene WH. The mixed logit model: the state of practice. Transportation. 2003 May;30:133-76.

31. McFadden D, Train K. Mixed MNL models for discrete response. J Appl Econ. 2000;15(5):447–70.

32. Hole AR. Mixed logit modeling in Stata--an overview. InUnited Kingdom Stata Users' Group Meetings 2013 2013 Sep 16 (No. 23). Stata Users Group.

33. van den Broek-Altenburg EM, Atherly AJ, Hess S, Benson J. The effect of unobserved preferences and race on vaccination hesitancy for COVID-19 vaccines: implications for health disparities. Journal of Managed Care & Specialty Pharmacy. 2021 Sep;27(9-a Suppl):S2-11.

34. Greene WH, Hensher DA, Rose J. Accounting for heterogeneity in the variance of unobserved effects in mixed logit models. Transport Res Part B Methodol. 2006;40(1):75–92.

35. van den Broek-Altenburg, E., et al., Exploring Heterogeneity in Moral Terminology in Palliative Care Consultations. BMC palliative care, 2020.

36. Yoo HI. lclogit2: An enhanced command to fit latent class conditional logit models. The Stata Journal. 2020 Jun;20(2):405-25.

37. Greene WH, Hensher DA. A latent class model for discrete choice analysis: contrasts with mixed logit. Transportation Research Part B: Methodological. 2003 Sep 1;37(8):681-98.

Qualitative research: p11 l232: As mentioned earlier, some paragraphs from the introduction can be added here

We have moved this text.

Qualitative research: p12 l259: “From the thematic analysis of the focus groups” – Which qualitative method (and/or software) was used? Did authors rely on any theoretical frame? In addition, this section is lacking real results. How many participants? What was their background and how were they recruited?

We added to the text to clarify the number of participants, the software that was used and how they were recruited.

Mixed Logit results:p14: Why was the number of iterations set to 100? Any rationale on that?

Thank you for this question as we realized the typo in the text, 100 should be 1000. The reason we used 1000 draws is that the literature suggests that, depending on the number of random parameters, stable mixed-logit estimation requires at least 1000 draws. We edited and added the justification.

Latent Class Analysis: Thank you. My comments from the first review have all been answered. Well done.

Thank you, we appreciate the comments and feedback.

Limitations: p16: I think a major limitation is the inclusion of only one university medical center. Due to that, the representativeness of the results must be questioned.

A discrete choice experiment / stated preferences research generally has a challenge with external validity since choice questions are asked in a hypothetical setting and study populations are usually not representative of specific choice settings. This is a known problem. For this reason, we had already written: “This study was performed within the hospital service area of a midsize academic medical center in a relatively rural area in the Northeast, so it is unclear how these results translate to patient preferences at the national level.” We have also refrained from making strong translations to a broader/national audience.

To further clarify this limitation, we cite the relevant literature and added:

“External validity remains a challenge for any DCE study even though, while an important component, it has been argued by others that the investigation of external validity should be much broader than a comparison of final outcomes.”

General: The manuscript should be rewritten/restructured to be more precise, especially in the methods section.

We have drastically rewritten en restructured the manuscript to address all the reviewer’s comments. We appreciate their comments.

Reviewer #2: I would like to thank the authors for their corrections.

However, I still have a few concerns I would like to see addressed.

Thank you, we very much appreciate your additional comments to improve the manuscript. We respond to each point raised below.

Note that the pages and lines I refer to are the ones in the track changes version.

Page 4, line 113 – “which has bene done in other fields” – typo, correct to “been”.

This word has been deleted in the new version of the manuscript.

Page 5, line 132 – “One option to measure hard to measure choice attributes is to use stated preferences (SP) data” – please avoid using more than two or three acronyms throughout the paper. In this case, you use SP so rarely that it probably didn’t even save more than two or three words.

Agreed. And this sentence has also been deleted from the text of the new version of the manuscript.

Page 8, line 203 to 205 – “In some cases, a study may contain more choice tasks than the researcher wants to ask respondents. In this case, a blocking experimental design can be used.” – I don’t think this sentence is very well written, to the point that it took me several readings to not understand the opposite of what you want to say. The case for using blocks is usually related to the need for a very big design and fear of cognitive burden. In those cases, researchers may split the design into 2 or more blocks so that each person answers a fewer amount of choice sets, avoiding cognitive burden and tiredness and therefore assuring the quality of the answers. I assume you want to say something within these lines, but that was not what was coming through.

Correct, that was the meaning of this sentence. We edited and it now reads: “In some cases, an efficient design includes many choice tasks. In this case, a blocking experimental design can be used to avoid too much of a cognitive burden for the respondent. Blocks are subsets of the choice questions, which are usually equally sized, that contain a limited number of choice questions for each respondent. In those cases, respondents are randomly assigned to a block and answer the choice questions in that block instead of the entire design.”

Page 8, line 208 to 210 – “In our study, respondents were asked 6 questions regarding X-ray services and 6 questions regarding Magnetic Resonance Imaging (MRI) services.” – Here I am confused. First, are these two block of the same design or two different designs? While you are using the same design for both, preferences may differ for each service, therefore, I would think you would have to randomly run the full design separately, which would count as two designs, even if they are equal. Second, is it correct to run the same design for both (e.g., shouldn’t some level values be different depending on the service?). Third, 6+6 is 12, but you say they had 14 choice sets. Where are the other two? Also, page 13, line 294 “134 were assigned to the arm X-ray group, 134 to the back MRI group.” Seems to imply they are just two different groups. I apologise if I completely misinterpreted what you wrote, but as of now, I think this part of the manuscript is not very clear.

You are completely right. Thank you for catching this. The text is wrong since this was a previous iteration of this study but we changed the approach. We have edited the text accordingly: “In our study, respondents were randomized to either 14 choice questions related to X-ray services or 14 questions releated to MR services. The vignette was different for the two DCE’s and some of the levels, such as costs, were also different. “

Page 9 – line 226 – Thank you for adding this table. Since the table in the choice sets needs to be simpler for visual simplicity reasons, I would like to know if respondents received instructions on the attributes and levels. This is way I stress that you should include the survey (or at least important parts of the survey) in appendix, for matters of clarity and for the reviewers to be able to properly evaluate the merits and limitations of the paper. Also, within table one, in the parking attribute, do you specify how much people need to pay in the level where they pay? Also, wouldn’t this be extremely correlated with the costs attribute? I see the point of the parking accessibility, since this implies other levels of stress in trying to park close to the place, but the cost is a little bit harder to understand. What is done is done, but it might deserve a comment from you. Another issue I detected is the Service attribute where I don’t know what these stars mean. Is it one star out of 5? What do they mean? Did respondents have access to this information (or maybe this is obvious to anyone living in the US?).

Thank you, we added the Scenario Introduction as an Appendix (supporting information). Since we created the survey in SurveyEngine, it is challenging to export the entire survey, but it will be available on request. We clarified using SurveyEngine for the online design in the methods section.

Thank you also for your comments about the parking costs and the star level attributes. We added text to clarify both. You could be correct regarding potential correlation between costs and parking. We addressed this in the limitations section: “Regarding the attribute of parking costs: this could be correlated with the cost attribute. We did ch

---

## [Decision Letter · Decision Letter 2]

17 Feb 2025

PONE-D-24-10399R2Patient Preferences for Diagnostic Imaging Services: Decentralize or not?PLOS ONE

Dear Dr. van den Broek-Altenburg,

Thank you for submitting your manuscript to PLOS ONE. After careful consideration, we feel that it has merit but does not fully meet PLOS ONE’s publication criteria as it currently stands. Therefore, we invite you to submit a revised version of the manuscript that addresses the points raised during the review process.

Both reviewers are generally satisfied with the revised version of your manuscript. However, one of them requests additional revisions to better address the potential cognitive burden. Moreover, the referee points out that the newly added sections contain several typos. Therefore, a thorough proofreading is highly recommended. I kindly ask you to take all the remaining concerns of the reviewer into consideration and to carefully adress them. Please submit your revised manuscript by Apr 03 2025 11:59PM. If you will need more time than this to complete your revisions, please reply to this message or contact the journal office at plosone@plos.org . Please include the following items when submitting your revised manuscript:

We look forward to receiving your revised manuscript.

Kind regards,

Matteo Lippi Bruni, PhD

Academic Editor

PLOS ONE

Journal Requirements:

Reviewers' comments:

Reviewer's Responses to Questions

**Comments to the Author**

1. If the authors have adequately addressed your comments raised in a previous round of review and you feel that this manuscript is now acceptable for publication, you may indicate that here to bypass the “Comments to the Author” section, enter your conflict of interest statement in the “Confidential to Editor” section, and submit your "Accept" recommendation.

Reviewer #1: (No Response)

Reviewer #2: (No Response)

2. Is the manuscript technically sound, and do the data support the conclusions?

Reviewer #1: (No Response)

Reviewer #2: Partly

3. Has the statistical analysis been performed appropriately and rigorously? 

Reviewer #1: (No Response)

Reviewer #2: Yes

4. Have the authors made all data underlying the findings in their manuscript fully available?

Reviewer #1: (No Response)

Reviewer #2: No

5. Is the manuscript presented in an intelligible fashion and written in standard English?

Reviewer #1: (No Response)

Reviewer #2: Yes

6. Review Comments to the Author

Reviewer #1: (No Response)

Reviewer #2: Thank you so much for your revision.

I do not have much else I want to see changed, so please try to address my final comments:

In this revision and the previous one, you added new text containing typos or with not optimally structured sentences, or even lacking clarity. Please re-read the bits that were added and adjust whenever needed.

For example, there is a typo at line 195, page 8 (track changes version): “The levels were partly define don concrete contributions”.

Another example, line 314, page 16: “The quotas that were agreed on included 5 percent of census on age by decade, gender, and region; 3 percent of census on race/ethnicity; and 50 percent above and below national median income of $65k/year.”. It is not very clear what you mean by 5 percent of census on age by decade, gender and region. Shouldn’t gender be around 50%/50%, for example?

Another one, on line 328, page 17: “We used a complex design for the DCE, although NGene showed that we had sufficient sample size for both X-ray and MRI.” – what is a complex design? Which method does NGene use? Enough sample size for the separate analysis on the X-ray group and MRI group of respondents?

Another one, line 576, page 29: “This could be due to parking being considered a “different kind of expense”, but we would leave out costs for parking out of the design in a next round of data collection.” – Future research is not relevant here. Just state the limitation and move on. Also, I wouldn’t talk about correlation tests because there is an obvious theoretical perfect correlation. Simple example, do you prefer to go to a GP consultation and pay 50 dollars + 10 dollars on parking or pay 60 dollars + 0 dollars on parking? They are the same because you pay 60 dollars anyway. The factor that may be relevant is whether parking is available. It is possible that respondents answered that part with another perspective. For instance, they may have associated free parking with how easy parking would be.

I would still like to see a comment on how likely it is that your design led to cognitive burden issues, because 14 choice sets with 3 alternatives each and 10 attributes is not easy task. Did you pre-test the design? Did people say it was ok? How long did they take to answer the survey? Please say something to inform the reader on whether this might have been a problem or not.

I would still like to see the survey. If you can’t find an easy way to export it, you should find another alternative (print page by page, copy paste question on another document, etc).

Thank you again for your revision!

7. PLOS authors have the option to publish the peer review history of their article (what does this mean? ). If published, this will include your full peer review and any attached files.

**Do you want your identity to be public for this peer review?** For information about this choice, including consent withdrawal, please see our Privacy Policy .

Reviewer #1: No

Reviewer #2: **Yes: ** Luis Filipe

---

## [Author Response · Author response to Decision Letter 3]

9 Mar 2025

Response to Reviewers.

Thank you for your additional comments. Please find our response to these issues.

In this revision and the previous one, you added new text containing typos or with not optimally structured sentences, or even lacking clarity. Please re-read the bits that were added and adjust whenever needed.

For example, there is a typo at line 195, page 8 (track changes version): “The levels were partly define don concrete contributions”.

We re-read the full manuscript and corrected all typos and sentences that needed to be edited.

Another example, line 314, page 16: “The quotas that were agreed on included 5 percent of census on age by decade, gender, and region; 3 percent of census on race/ethnicity; and 50 percent above and below national median income of $65k/year.”. It is not very clear what you mean by 5 percent of census on age by decade, gender and region. Shouldn’t gender be around 50%/50%, for example?

The percentages refer to the Census method of covering a particular percentage of the total population. We left this out to avoid confusion.

Another one, on line 328, page 17: “We used a complex design for the DCE, although NGene showed that we had sufficient sample size for both X-ray and MRI.” – what is a complex design? Which method does NGene use? Enough sample size for the separate analysis on the X-ray group and MRI group of respondents?

We did mean “complex” but a design with many attributes and levels. We created an efficient design with Ngene to reduce the dimensions of the matrix and getting optimal information. A design is considered more efficient if it can produce more efficient data in the sense that more reliable parameter estimates can be achieved with an equal or lower sample size. This process is explained on page 10.

Another one, line 576, page 29: “This could be due to parking being considered a “different kind of expense”, but we would leave out costs for parking out of the design in a next round of data collection.” – Future research is not relevant here. Just state the limitation and move on. Also, I wouldn’t talk about correlation tests because there is an obvious theoretical perfect correlation. Simple example, do you prefer to go to a GP consultation and pay 50 dollars + 10 dollars on parking or pay 60 dollars + 0 dollars on parking? They are the same because you pay 60 dollars anyway. The factor that may be relevant is whether parking is available. It is possible that respondents answered that part with another perspective. For instance, they may have associated free parking with how easy parking would be.

Agreed. We deleted “We did check for correlation and do not find the two coefficients to be correlated. This could be due to parking being considered a “different kind of expense”, but we would leave out costs for parking out of the design in a next round of data collection.”

I would still like to see a comment on how likely it is that your design led to cognitive burden issues, because 14 choice sets with 3 alternatives each and 10 attributes is not easy task. Did you pre-test the design? Did people say it was ok? How long did they take to answer the survey? Please say something to inform the reader on whether this might have been a problem or not.

We added a clarification to the limitations section.

I would still like to see the survey. If you can’t find an easy way to export it, you should find another alternative (print page by page, copy paste question on another document, etc).

We uploaded the html version of the survey text as a supplemental file.

---

## [Editor Report · Decision Letter 3]

25 Mar 2025

Patient Preferences for Diagnostic Imaging Services: Decentralize or not?

PONE-D-24-10399R3

Dear Dr. van den Broek-Altenburg,

We’re pleased to inform you that your manuscript has been judged scientifically suitable for publication and will be formally accepted for publication once it meets all outstanding technical requirements.

Kind regards,

Matteo Lippi Bruni, PhD

Academic Editor

PLOS ONE
---

## [Editor Report · Acceptance letter]

PONE-D-24-10399R3

PLOS ONE

Dear Dr. van den Broek-Altenburg,

I'm pleased to inform you that your manuscript has been deemed suitable for publication in PLOS ONE. Congratulations! Your manuscript is now being handed over to our production team.

Kind regards,

on behalf of

Dr. Matteo Lippi Bruni

Academic Editor

PLOS ONE